# Trading off fiscal budget adherence and child protection

**Petra Gram Cavalca**, **Mette Ejrnæs**, **Mette Gørtz** *

Department of Economics, University of Copenhagen and CEBI, Copenhagen K, Denmark

* mette.gortz@econ.ku.dk

**Data Availability Statement:** The data used in the paper are administrative Danish register data, which are kept in a secure server maintained by Statistics Denmark. The data can, however, be accessed remotely from within Danish universities

## Abstract

Many countries delegate a substantial part of social service decisions to local administrative levels, while federal laws provide the overall framework for service levels. Strict regulations to reduce budget overruns may however leave local governments with a potential trade-off between adhering to fiscal budgets and supplying critical welfare services as e.g. programs to protect vulnerable children. We investigate if budgetary constraints influence child protection decisions using high-quality register data. We show that the introduction of fiscal sanctions to improve budget adherence contributed to a sharp decline in budget overruns on child protective services by reducing the number of children in out-of-home care. Our results further show that monthly variation in budget adherence within a fiscal year affects the probability of a placement in out-of-home care for children in need of help towards the end of a fiscal year. We estimate that a budget overrun of 10 percentage points by mid-year leads to a 1.2 percent reduction in the number of children in care over the remaining part of the fiscal year. Municipalities reduced child protection expenditure by choosing cheaper types of care and ending placement for children in out-of-home care, particularly for children turning 18. Our paper contributes to the literature on fiscal federalism by documenting the trade-off between managing public expenditure and providing safety and equal opportunity for vulnerable children. We thus highlight that enforcing strict budget adherence may be in conflict with social policy goals. Our results raise an important discussion about centralization versus delegation of critical public services.

## Introduction

Governments all over the world struggle to curb public expenditure. Tight budgets combined with severe economic sanctions for budget overruns have led to fiscal restraint at the local administrative level across all domains of public services in the last decade [1–3]. At the macro level, budgetary austerity has in some countries, like e.g. Denmark, led to improvements of public budgets, providing the foundation for a reserve buffer. An important part of public budget savings can be ascribed to budget reductions at the regional and municipal level, as a substantial proportion of public budgets are delegated to local government levels. However, imposing strict budget adherence can leave local governments with a trade-off between providing critical services such as child protection and overrunning the budget.

and research institutions. For more information on access to Danish register data, we refer to Statistics Denmark's webpage https://www.dst.dk/en/TilSalg/Forskningsservice. The authors will provide all programs and instructions to any researcher who should wish to replicate our paper. All authors have had full access to the data used in the paper.

**Funding:** All three authors of the project received funding from: - Trygfonden, https://www.tryghed.dk/, grant no. "2013 pulje - nr. 12". - The Danish National Research Foundation, https://dg.dk/en/, through its grant no. DNRF-134 to CEBI, Center for Economic Behavior and Inequality. The funders had no role in study design, data collection and analysis, decision to publish, or preparation of the manuscript.

**Competing interests:** The authors have declared that no competing interests exist.

The literature on fiscal federalism highlights several advantages to delegating public services to a local level [4–9]. *First*, it may be easier to collect information on the public needs at the local level. Child protective services often rely on referrals from the local community, such as neighbors or schoolteachers, to identify children in need of help. *Second*, there may be an advantage to locally organized services, for example in supporting parents or recruiting a foster family close to the biological parents' home. *Third*, if local governments handle the provision of public goods, they can more easily adapt to the preferences of local citizens. An argument referred to as the "Tiebout hypothesis" [10–12] states that citizens can signal their preference for public goods by moving to a local jurisdiction matching their preferences.

Recent literature questions the basic rationale behind decentralizing public services from the perspective of three concerns: efficiency, equity, and accountability [13]. Decentralization offers specific challenges for public expenditures related to for example health and education [11]. It has been shown that the demographic composition of local areas determines local public expenditure levels. For example [14], document that the percentage of elderly adults in a school district is negatively related to the amount of support for public schooling.

While fiscal budgets stipulate the overall frame for public expenditure at the national and local administrative level, the provision of some public services is also regulated by law. Local governments may thus face a trade-off between showing fiscal restraint and supplying critical, law-mandated welfare services to vulnerable citizens. Child protection is a particularly important and relevant example of this trade-off. Child maltreatment has severe negative consequences for child health and well-being [15–20]. Each year almost 1 percent of US children spend time in care and 6 percent of US children have been placed in foster care at least once before turning 18 [21,22]. In most Western countries, federal child protection laws aim to improve equity in conditions and secure basic rights and safety for at-risk children. In practice, the administration of these laws is usually delegated to a local level such as municipalities or counties. Even small deviations from the expected number of children who receive costly interventions such as out-of-home care can lead to a budget overrun in a small municipality. This can leave the local administration in a conflict between budgetary adherence and the statutory responsibility to take action when a child needs help.

The research question we answer in this paper is whether variation in local budgetary pressure—*within* a fiscal year—influences child protection decisions. We investigate the question empirically by examining whether municipalities that have spent a larger than expected fraction of their budget on child protection in the first part of the year subsequently reduce the number of children in care. We use a rich set of individual-level data to examine the mechanisms through which municipalities reduce expenses on child protection. We explore the causal effects of a reform, effective since 2011, that implemented sanctions on individual municipalities for overrunning budgets. To set our estimation results into perspective, we calculate that municipalities that have spent more than 60 percent of their budget by the middle of a fiscal year reduce the number of children in out-of-home care by 1.2 percent by the end of the year.

To test the robustness of our results, we exploit a reform that incentivized municipalities' budget adherence. We show that the introduction of fiscal sanctions contributed, although only moderately, to a decline in budget overruns on spending allocated to vulnerable children and, unintentionally, affected the provision of child protective services. Municipalities reduced child protection expenditures by choosing cheaper types of out-of-home care and by being more likely to end out-of-home care, particularly after a child's 18th birthday, after which the municipality is no longer legally obligated to provide child protective services.

Moreover, we perform a placebo test to rule out that the results we find are due to mean reversion or deliberate timing of municipal activities over the year.

Our paper contributes to the literature on fiscal federalism by documenting a trade-off between, on the one hand, providing effective and credible measures to curb local government expenditure and, on the other hand, ensuring safety and equal opportunities for vulnerable children. While variation *across* municipalities in the services provided is well-known from other studies [23], we document that *within-municipality* variation in expenditure over the fiscal year has an additional effect on the services offered to at-risk families. Our primary contribution is to show that the *timing*—within a fiscal year—of municipal expenditures affects the provision of child protective services.

Our results highlight important side effects of imposing strict budget adherence at the local level and contributes to a policy discussion about centralization versus delegation of critical public services. To design an effective public sector, an understanding of how local governments deal with conflicting requirements is key and can help delegate public services to the most appropriate administrative level. It is important to know if policies designed to uphold fiscal stability have unintended consequences, particularly if it affects vital welfare tasks such as child protection. Whereas the traditional advantages of decentralization such as local information and local organization clearly apply in the case of child protection, we argue that it is unclear whether it is advisable to adapt the provision of child protection to local demand. Child protection receives less attention in the local public debate than large welfare goods such as health and education, and it is rarely high on the political agenda for local elections. Demand for child protection from citizens not directly involved may suffer from a lack of information, as is often the case in areas characterized by stigma, and only a small fraction of the population is directly involved with child protective services. Furthermore, the biological parents of at-risk children may prefer no public intervention, and the children do not have the power to 'vote with their feet' as the Tiebout hypothesis would suggest. For a theoretical discussion of utility functions in the Tiebout model, see [24]. If society's main concern is children's safety and well-being, adapting the provision of child protective services to local preferences may thus be problematic.

## Background and institutional setup

Financial stability of local government budgets is at the center of fiscal policy in many Western countries. Legislation implemented in 2011 in Denmark imposed expenditure ceilings on municipal budgets and spending. The expenditure ceilings are enforced through economic sanctions if the municipal budget exceeds the centrally mandated budget or if municipal expenditures exceed the budget. Individual municipalities that overrun their annual budget are required to pay 60 percent of the sanction imposed on the municipalities as a whole [2,25]. Since municipal budgets have complied with the expenditure ceilings it has not so far been necessary to use sanctions as a disciplinary instrument.

Municipalities make separate budgets for all their activities, including spending on out-of-home care, which amounted to about 2.4 percent of total municipal expenditures in 2016. The municipalities' budgets on out-of-home placement increased from 2007 to 2009, followed by a reduction after 2010, coinciding with the financial crisis and the national efforts to increase financial stability. Budgets vary substantially across municipalities, with budgets around 6 million Euro for municipalities at the 25[th] percentile, and budgets around 13 million Euro at the 75[th] percentile, depending, among other things, on the size of the municipality (see S3 Fig in the S1 Appendix).

The Danish child protection law stipulates that the main goal of child protective services is to support at-risk children to "obtain the same opportunity for personal development, health and an independent adult life as their peers" (Law on Social Services—*Serviceloven*, Ch. 11,

§46). Municipalities are responsible for the care of vulnerable children in Denmark and may assign a range of interventions, from various types of preventive action to out-of-home care as the most drastic intervention. Out-of-home care is intended to be a temporary intervention and the final goal is reunification with the biological parents. Similar to numbers in the US, almost 1 percent of Danish children aged 0–17 spend time in out-of-home care each year [26,27]. Since 2007, the 98 Danish municipalities have had full fiscal responsibility for at-risk children and it is the municipality's decision to place a child in out-of-home care. Parents can appeal the municipality's decision to the National Social Appeals Board [28]. Child protection mainly concerns children aged 0–17, however the municipality can extend the out-of-home placement up to the age of 22. Around two-thirds of children in care live in family foster care and a third live in institutional care (own calculations on register data, see Data section for details). For children exiting care, the average length of care was around four years per placement. Less than one-third of children in out-of-home care return to their biological parents before age 18, and the rest "age out" of care at age 18 or over. Many children transition from one type of care into another, and children experience an average of 1.4 placements. According to a recent survey based study among caseworkers in Copenhagen [27], the most common reasons for placing a child in out-of-home care is parental neglect (50 percent) and child externalizing behavior and social adjustment issues (33 percent). Less frequent reasons are violence or threats of violence (10 percent) or sexual abuse (2 percent).

Out-of-home care is generally very costly, but costs vary considerably by type of placement. The average cost for a child in institutional care amounts to more than 150,000 Euro annually, while the cost of a foster family is around 68,000 Euro annually (details on prices for child protection programs in S1 Table in S1 Appendix). For small municipalities, placing one additional child in out-of-home care is likely to pose a serious threat to budgetary compliance. Three additional (unexpected) children in institutional care would result in a budget overrun of almost 5 percent for the median municipality and more than 9 percent for the 25 percent smallest municipalities.

Fig 1 shows the average budget overrun on the budget for out-of-home care and the total budget for Danish municipalities in our sample period. While municipalities were heavily overspending on out-of-home placements from 2007 to 2010, overspending dropped from around 13 percent at its peak in 2009 to around 2 percent in 2011 after the introduction of sanctions on overspending.

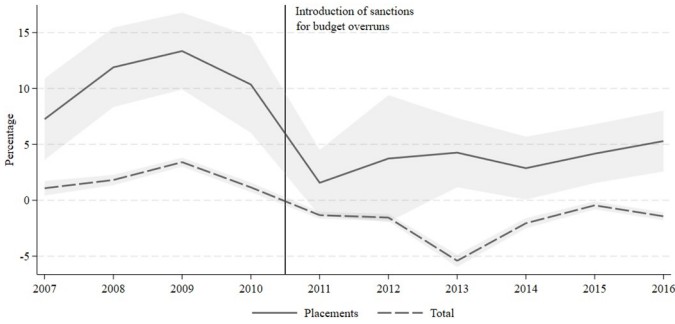

**Fig 1. Budget overruns on out-of-home care.** The graphs shows the average of annual actual expenses—expenses in the budget relative to the budget. The solid line is for expenses on out-of-home placement whereas the dashed line is for all municipality expenses. The vertical line indicates the introduction of sanctions against budget overrun. The shaded area indicates the 95 percent confidence interval. Source: Own calculations based on data from Statistics Denmark, Statistikbanken, Table BUDK53 and REGK53.

Budget overruns increased from 2007 to 2009 on most municipal activities, followed by deficit reductions after 2010, but the changes over time were much stronger for out-of-home care than for the other areas (see details in the S1 Appendix, S5 Fig in S1 Appendix). The averages mask considerable heterogeneity in over- and underspending across municipalities. Budget overruns occurred in municipalities in various parts of the country before 2010, while they were much less widespread after 2011 (see map of municipalities with budget overruns in S4 Fig in S1 Appendix). Almost all municipalities (95 percent) overspent on their out-of-home care budgets in at least three of the ten years in our sample period.

## Data and methods

### Data

Our empirical analyses use unique Danish register data, a longitudinal dataset with individual-level information on socioeconomic characteristics including family status, education, income, government transfers and child protective services such as preventive action, out-of-home care spells and type of care. The sample consists of all children and youth receiving preventive action or out-of-home care during the years 2007 to 2016. Data cover 97 Danish municipalities (excluding the capital Copenhagen due to its size). The median Danish municipality had 120 children in out-of-home care in 2007. This number had decreased to 110 in 2016.

A crucial measure for our analyses is the municipalities' monthly expenditure for out-of-home care. Data on municipalities' budgets and accounts are annual, so we construct a measure of monthly spending using individual-level data. The individual register information allows us to calculate exactly how many children were in out-of-home care each month in each municipality. Furthermore, we can divide these "care months" according to the type of care. Based on the number of "care months" of each type of care and the average prices per care type, we impute a measure of monthly expenditure for municipalities (see more details in the S1 Appendix). We construct a monthly variable, the "budget share", which measures the cumulative monthly expenditure as a fraction of the planned annual budget. As an example, the budget share on March 1st is the sum of imputed expenses for January and February divided by the annual budget for that year. The budget share will be larger than one if a municipality's cumulative expenditures for out-of-home care in a given month exceeds the total annual budget. Fig 2 shows boxplots of the imputed budget share at the first day of the month over the fiscal year for all municipalities in the period 2007 to 2016. The budget share is by construction zero by January 1 for all municipalities. By October 1, the median municipality had spent about 80 percent of the annual budget, while some municipalities had spent substantially more and a few had already spent the entire annual budget. The divergence across municipalities in over- or underspending increases over the fiscal year.

Our identification strategy consists of two approaches to estimate the effect of budget shares on out-of-home placement decisions. The first approach uses municipal level data and allows us to quantify the total effect of budget shares on the number of children in out-of-home care. The second approach uses individual-level data to investigate the mechanisms to reduce expenditures on out-of-home care. Both analyses rely on a comparison of high- and low-spending municipalities across time. Consider a simplified illustration of our approach in which there are two municipalities, A and B. We imagine that municipality A at the beginning of July that year had spent more than 60 percent of its budget on out-of-home care. Hence, municipality A is a high-spending municipality in that given year. Low-spending municipality B had spent only 50 percent of its out-of-home care budget by July 1 that same year. While municipality A had to cut expenditures for out-of-home care in the last six months of the year to stay within the budget, municipality B did not need to adjust its expenditures. Imagine that

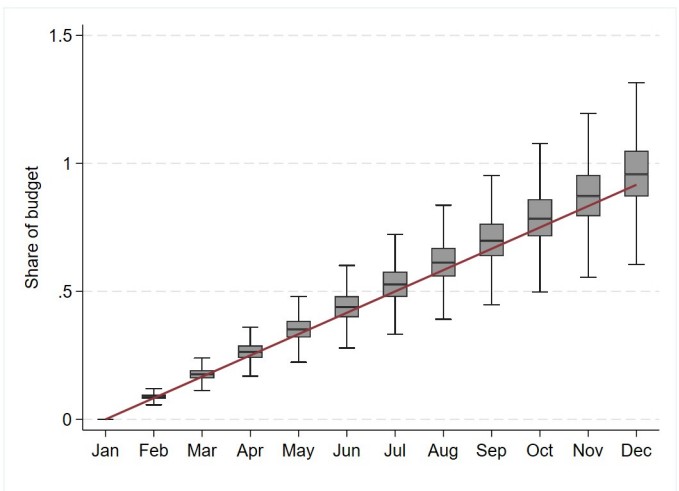

**Fig 2. Municipalities' expenditure as share of the budget, 2007–2016.** Budget share is measured at the beginning of the month. For January, the budget share is always zero. For February, the budget share is defined as the proportion of the total budget for year $t$ that was used in January. For March the budget share is calculated as the share of the January and the February expenditure out of the total budget, etc. Finally, for December, the budget share is defined as expenditure share of the months January to November in that year. The red line indicates the expected budget share, if the expenses were equally distributed across the months.

while municipality A had spent more than 60 percent of its out-of-home care budget by July 1 in year 1, it had only spent 50 percent of the budget in year 2. Municipality A was high-spending in year 1, but it was low-spending in year 2. Our analysis also exploits within-municipality variation across years. Using both within and between municipality variation allows us to account for both municipality fixed effects and calendar time fixed effects. It is important to note that although this stylized example considers a budget share threshold of 60 percent on July 1, our approach measures the effect of a marginal increase in the budget share at any value and for all months of the year. We will now describe each of the two approaches in more detail.

### Municipal-level analysis

The *municipality-level* analysis estimates the total effect of budget shares on the number of children in out-of-home care. The analysis builds on an error correction model, where the number of children in out-of-home care, $y_{kt}$, in municipality $k$ in month $t$ depends on the long-run municipality specific mean, $\mu_k$, and the budget share.

$$\Delta \log y_{kt} = \alpha_1 (\mu_k - \log y_{kt-1}) + \alpha_2 (Z_t - Budgetshare_{kt-1}) + \theta_t + \varepsilon_{kt} \qquad (1)$$

where *Budgetshare* is defined as the share of the budget for out-of-home care that has been spent from January 1 until month $t$-1. We interpret the long-run mean as the expected number of children in need of out-of-home care in the municipality. $Z_t$ measures the expected seasonal profile of the budget share, and $\theta_t$ contains year and month dummies. We hypothesize that the number of children in care adjusts to the long-run mean, $\mu_k$, and to the deviation between actual expenditure and the expected seasonal profile. We expect $\alpha_1 > 0$, suggesting that the number of children in care converges to the long-run mean. We also expect $\alpha_2 > 0$ if deviations from the expected spending patterns lead to an adjustment in the number of children in

care. Rearranging and simplifying (1), we estimate the following regression model

$$\Delta log y_{kt} = \alpha_0 - \alpha_1 log y_{kt-1} - \alpha_2 Budgetshare_{kt-1} + \tau_k + \varphi_t + \varepsilon_{kt}, \qquad (2)$$

where $\tau_k$ contains municipality dummies and $\varphi_t = \alpha_2 Z_t + \theta_t$ contains year and month dummies. The municipality-level analysis exploits variation over time within municipalities to estimate the effect of an increase in the monthly budget share on the number of children in out-of-home care.

## Individual-level analysis

The *individual-level* analysis provides information on which margins municipalities reduce their expenditures. There are three relevant outcome margins to consider; the municipality can reduce expenditures by either ending placement for children already in care, placing fewer children in care or by choosing a less expensive placement for children who are placed in care. We estimate the effect of budget share on probabilities of five separate outcomes: 1) ending out-of-home placement for children younger than 18 years old, 2) ending out-of-home placement for children above 18, 3) initiating a new out-of-home placement for children already receiving a preventive measure, 4) initiating a preventive care measure rather than an out-of-home placement, and 5) choosing a less expensive type of care for all children placed in out-of-home care. For details on sample selection for each outcome, see S2 Table in S1 Appendix. We estimate the relationship between the probability of each individual outcome and monthly budget shares using a logit specification:

$$y_{it}^* = \gamma \cdot Budgetshare_{kit-1} + X_{it}\beta + \mu_k + \theta_t + \varepsilon_{it}, \qquad (3)$$

where $y_{it}^*$ is the latent outcome. *Budgetshare* is defined as the share of the annual municipal budget for out-of-home care spent by month $t$-1. $X_{it}$ includes socioeconomic characteristics, such as the child's gender, age, birthweight, dummy for ethnic minority background, whether mother or father is not on record, and maternal characteristics (age, income, labor market status, education and marital status). We include a full set of municipality dummies ($\mu_k$), and year and month effects ($\theta_t$). Standard errors are clustered at the municipality-level. The individual-level analysis exploits variation within municipalities and across time to estimate the effect of an increase in the monthly budget share on the individual probability of ending out-of-home placement, initiating a new placement or choosing a cheaper placement type. This approach allows us to investigate how municipalities adjust the number of children in care in response to a higher budget share.

## Threats to identification

A potential threat to the identification strategy comes from mean reversion. If some municipalities deliberately organize their work over the year to place relatively many children in out-of-home care in the first half of the year for administrative reasons or seasonal variation, we may observe a high budget share by July 1 followed by fewer placements in the second half of the year. This would show up in our results in the same way as an effect of budget overrun on the number of children placed in care. We perform three different investigations to rule out that our results are driven by mean reversion.

*First*, our municipality-level regression explicitly controls for mean reversion by estimating an extended error correction model for the change in number of children in care. We allow each municipality to converge to their own long-run mean for the number of children in care. In this way, we explicitly test for mean reversion by including the lagged number of children in care in our estimation.

*Second*, the reform in 2011 encouraged municipalities to pay more attention to the budget after 2011. If municipalities reacted more strongly to budget overruns after 2011, this would strengthen the interpretation that municipalities reduce out-of-home placements towards the end of the year due to the risk of budget overruns. To explore the effect of the 2011 sanctions, we interact the lagged *Budgetshare* with a dummy for years after 2011.

*Third*, we perform a placebo test where we assume a hypothetical fiscal year running from July to June. If potential effects on out-of-home care were the result of seasonal patterns or deliberate organization of activities in municipalities, we would also find an effect of budget shares on placement rates using the alternative fiscal year running from July to June.

## Results

### Municipal-level analysis

We start by showing descriptive evidence on how municipalities adjusted their spending in the second half of a budget year following a budget overrun in the first half of the year. We split the municipalities into two groups according to their spending patterns in the first six months of the fiscal year. The high-spending group consists of municipalities that spent more than 60 percent of their total annual budget by July 1. The low-spending group consists of municipalities that spent less than 60 percent by July 1. We also looked at the difference between high- and low-spending municipalities using a different threshold value, and this does not change the conclusion. The descriptive evidence illustrates the effect we are interested in, but this effect is not necessarily causal. To estimate the causal effect of a marginal increase in budget share, we turn to the more rigorous municipality-level analysis. Fig 3(a) depicts the monthly change in the number of children in out-of-home care for the second half of the fiscal year, i.e., from July to December for low and high-spending municipalities, for the periods before and after 2011, the year of the municipal budget reform. Fig 3(a) clearly documents two findings. First, there is a reduction in placements in out-of-home care after 2011 for both high- and low-spending municipalities. Second, the number of placements dropped significantly more for high-spending municipalities compared with low-spending municipalities after 2011. In total, the number of children in out-of-home care dropped, on average, 4.5 percent from July to December (a monthly decrease of around 0.75 percent).

We examine the impact of the budget share in the error correction model in Eq (1) using monthly data at the municipality level. Estimation results are shown in Table 1, column 1. There is a negative and significant effect of the lagged log of number of children in a municipality in period $t$-1 on the change in the number of children in out-of-home care from period $t$-1 to $t$. This indicates that there is mean reversion to a municipality specific "long-run" mean of number of children in care. Our main interest lies in the effect of the budget share in period $t$-1 on the change in log number of children in out-of-home care in period $t$. We estimate that if the budget share is 10 percentage points higher than expected, the number of children in care decreases by 0.2 percent per month. This is equivalent to a 1.2 percent reduction in the number of children in out-of-home care over half a year. The effects are not significantly different when comparing before and after 2011 (see column 2 in Table 1) in the municipal-level analysis. Column 3 in Table 1 shows the result from estimating Eq (2) using a placebo fiscal year; we return to this placebo test in the end of the section. We also consider if the budget share affects the number of children receiving preventive actions such as e.g. support for the family. These interventions are often used to avoid out-of-home placement. The analysis shows (see column 4, Table 1) that municipalities at risk of budget overrun lower the number of children receiving preventive actions after 2011 but not before 2011. If the budget share is,

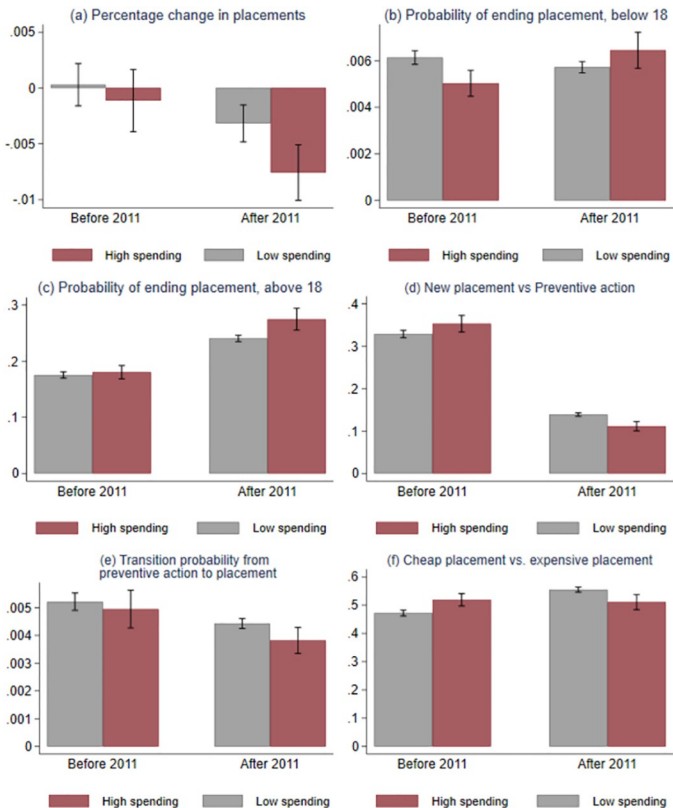

**Fig 3. The monthly probability of ending and initiating out-of-home placements.** High (Low) spending municipalities in year $t$ are defined as municipalities that in year $t$ have spent more (less) than 60 percent of the annual budget by July $1^{st}$. Panel (a): Outcome is changes in log number of children in care. Panel (b) and (c): Outcome variable is monthly probability of ending an out-of-home placement for children below (above) 18. Panel (d): Outcome variable is monthly probability of initiating an out-of-home placement. Panel (e): Outcome is probability for a child (below 18) who is not currently receiving preventive actions and the left panel is for a child that receive preventive actions. Panel (f): Outcome is probability of choosing a "cheap" placement instead of an expensive placement. Confidence bands at 0.05 level.

e.g., 10 percentage points higher than expected, the number of children receiving preventive actions decreases by 0.9 percent the following month after 2011.

## Individual-level analysis

We now examine the mechanisms through which the municipality adjusts its expenditure on child protection using individual-level data. We consider five different outcomes: ending a placement for children aged below 18, ending a placement for children above 18, initiating a new placement, initiating a placement for children who receive preventive actions and choosing a less expensive placement versus an expensive placement (see details in S2 Table in S1 Appendix). The less expensive care is primarily foster families while the expensive care is institutional care. Each of the five outcomes represents a margin where the municipality can adjust their expenditure on child protection. Fig 3(b)–3(f) shows descriptive evidence on the margins. The figure shows differences in the probability of each of the five outcomes in the last six months of a fiscal year (July to December) for high- and low-spending municipalities, before and after 2011. The probability of ending a placement in July to December is conditional on spending in January to June. Fig 3(b) and 3(c) show that the probability of ending a placement

**Table 1. Estimation results, municipality-level analysis.**

| | (1) | (2) | (3) | (4) |
|---|---|---|---|---|
| Outcome ($y_{kt}$) | | Num. of children in care | | Num. of children receiving preventive actions |
| | **Baseline** | **Estimation with interaction** | **Placebo Estimation** | **Estimation with interaction** |
| $\log(y_{kt-1})$ | -0.023*** | -0.023*** | -0.034*** | -0.052*** |
| | (0.005) | (0.005) | (0.006) | (0.006) |
| Lag Budget share | -0.017* | -0.017* | 0.009 | 0.011 |
| | (0.007) | (0.006) | (0.005) | (0.006) |
| Lag Budget share | | 0.003 | | -0.020*** |
| × D2011 | | (0.002) | | (0.002) |
| $\log(y_{kt-1})$ | | 0.000 | | -0.002 |
| × D2011 | | (0.001) | | (0.002) |
| Constant | 0.114*** | 0.114*** | 0.166*** | 0.337*** |
| | (0.026) | (0.024) | (0.028) | (0.031) |
| Year dummies (9) | Yes | Yes | Yes | Yes |
| Month dummies (11) | Yes | Yes | Yes | Yes |
| Muni. Dummies (96) | Yes | Yes | Yes | Yes |
| N | 11,531 | 11,531 | 10,464 | 11,531 |
| $R^2$ | 0.034 | 0.035 | 0.038 | 0.175 |

The estimation results refer to estimation of Eq (2) with the dependent variable being $\Delta \log y_{kt}$. The model is estimated using monthly data for the period 2007–2016 at the municipality level. The data set consists of 97 municipalities (The municipality of Copenhagen is excluded). In column (1)-(3) the outcome is number children in care and in column (4) the outcome is number of children receiving preventive actions. Colum (3) contains a placebo test where the budget share is replaced by a hypothetical budget share (for more details see S1 Appendix). D2011 is a dummy for the period 2011–2016. The estimations include 11 month dummies, 9 year dummies and 96 municipality fixed effects. Standard errors in brackets are clustered at the municipality level.

*, **, *** indicate significance at 0.05, 0.01 and 0.001 level.

increased, for children under 18 (Fig 3(b)) and children above 18 (Fig 3(c)) after 2011. The largest difference is for youngsters above 18. The probability of out-of-home care reduced after 2011 for out-of-home care versus preventive action (Fig 3(d)) and for the transition from preventive action to out-of-home placement (Fig 3(e)). We also observe a substitution away from the more expensive institutional care to the less expensive foster family care, especially after 2011 (see Fig 3(f)). High-spending municipalities were more likely to end placements after 2011, especially for young adults, and they were less likely to initiate new placements than before 2011.

We examine the effect of the budget share on the five outcomes in a logit estimation, with year, month and municipality dummies and a number of individual socioeconomic characteristics, such as child gender, age, and birthweight, marital status of parents, and mother's age, education and employment (detailed estimation output in S1 Appendix, S4 Table in S1 Appendix).

## Marginal effects

Fig 4 illustrates the estimated marginal effects by budget share on July 1 from model (3) for the three outcomes with significantly estimated effects: ending out-of-home care for children below and above 18 and choosing a less expensive type of out-of-home care. Fig 4(a) shows the likelihood of reunification for children in care as a function of the share of the budget spent by the municipality by July 1. The figure indicates that the monthly probability of reunification is 0.65 percent if a municipality has spent 50 percent of its budget by July 1, while it was 0.7

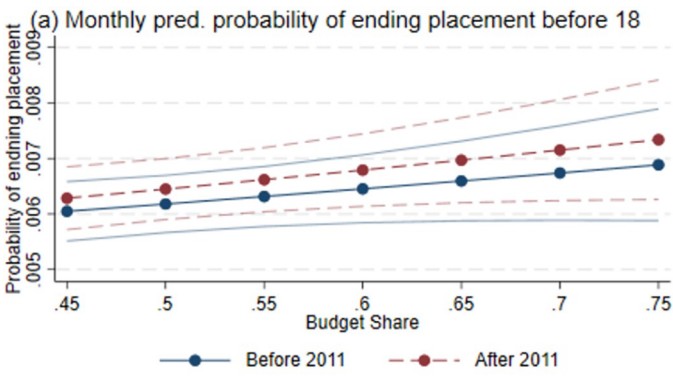

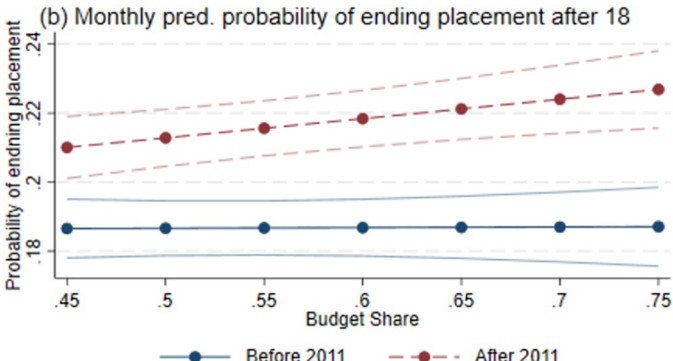

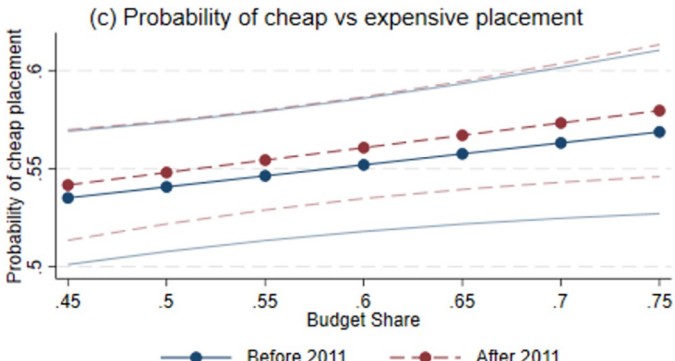

**Fig 4. Predicted probabilities by budget share, individual-level regressions.** The Figures show the marginal effects of the budget share on the individual outcomes. Panel (a) shows the predicted monthly probability of ending a placement for children in care below 18. Panel (b) shows the predicted monthly probability of ending a placement for children above 18 in care. Panel (c) shows the predicted probability for a cheap out-of-home placement (e.g. foster families) instead of an expensive (institutional care). The predictions are based on estimates of the logit specification in Eq (3). The predicted probabilities are calculated from July. The logit model is estimated using individual-level data from the period 2007-2016. The dashed lines indicate the 95 percent confidence interval.

percent if the municipality had spent 60 percent of its budget. Fig 4(b) shows that the probability of ending out-of-home care placement for children turning 18 was significantly higher after 2011 than before, for all levels of budget shares at July 1. To quantify these effects, municipalities that had, after 2011, spent 10 percentage points more of their budget by mid-year (e.g., having spent 60 rather than 50 percent), had a 0.5 percentage point higher propensity to end placement for over 18-year olds. When compared to an average probability of ending out-of-

home care of around 21 percent for children who turn 18, this is equivalent to a 2–3 percent increase in the probability of ending out-of-home care for over 18-year olds. For the choice between a less expensive and an expensive placement (Fig 4(c)), we see that a higher budget share increases the probability of the less expensive placement, but the difference is small and statistically insignificant when comparing before and after 2011. After 2011, the effect of a higher budget share is stronger for all outcomes. For all levels of budget shares, the probability of ending placement after age 18 is significantly higher after 2011 as compared to the situation before the 2011-reform (Fig 4(b)).

### Heterogeneous effects and robustness

We find suggestive evidence of heterogeneity in how sensitive municipalities are to budget concerns (see S5 Table in S1 Appendix). The individual-level analyses show that budget concerns have a smaller impact after on out-of-home care decisions in municipalities where the majority in the municipal council consists of parties on the center-left (this result is borderline significant) or if less than 30 percent of the municipality council are women. We also find that municipalities that have sizable debt are more sensitive to the risk of budget overrun before 2011 and that municipalities are less sensitive to risk of budget overrun in election years before 2011. The last result is surprising, but may be due to the fact that child protection policies are rarely a topic in local elections where politicians may prioritize topics of greater importance to the majority. If the majority in the local council belongs to the same political bloc as the prime minister the municipality is more sensitive to budget concerns. We looked for heterogeneous effects in other dimensions: population size, number of placements by January 1$^{st}$ 2007, a measure of needs of local residents, educational composition, fraction of single parents, average income (for a detailed description of the all measures see the S5 Table in S1 Appendix). Neither of the measures seem to matter for the size of the effect. The municipality-level analysis shows no indication of heterogeneous effects (see S6 Table in S1 Appendix).

To test the robustness of our results, in particular whether our results could be driven by the way municipalities organize their work with at-risk children over the fiscal year (mean reversion), we run our analysis using a placebo budget share. We hypothesize that the fiscal year runs from July 1 to June 30 the following year (see S7 Fig in S1 Appendix). We define a placebo annual budget as the average budget for two consecutive calendar years, and we construct the placebo budget share for each month from the monthly expenditures and the placebo annual budget as described in the data section. We repeated our analyses with the placebo budget share instead of the real budget share. The placebo test indicated no effect of budget overruns in our "placebo" fiscal year. This confirms that our main results are not driven by mean reversion (see Table 1, column 3 for the municipality-level estimations, and S8 Fig in S1 Appendix for the individual-level estimations).

### Discussion and conclusion

Throughout the world, governments struggle to curb public expenditure at all administrative levels. In Denmark, this has led the government to impose fiscal sanctions on local municipalities who overrun their annual budgets. As a large share of public expenditure is spent at local levels—municipalities and regions—budgetary restraint at the local levels is important. In this paper, we show that such fiscal sanctions can have substantial effects on essential welfare services that are designed to protect some of the most vulnerable citizens, namely children at risk of neglect and maltreatment.

While other studies have studied the variation *across* municipalities in service levels [23], we document that *within-municipality* variation in expenditure over the fiscal year impacts

services provided to families. Thus, an at-risk child may receive differential treatment depending on whether information about the case is brought up in December or January. Using individual-level administrative register data, we show that municipalities at risk of overrunning their budget reduced the number of children in out-of-home care. We find that a 10 percentage point increase in the budget share by July decreased the number of children in out-of-home care by 1.2 percent by the end of the fiscal year. The detailed empirical analysis on individual-level data shows that municipalities primarily reduced the number of children in care by ending care for children in out-of-home care and by using less expensive types of out-of-home placement. These results suggest that local policy makers face a trade-off between meeting fiscal targets and offering public services mandated by law, such as policies to assist vulnerable children. The estimated effects on out-of-home care placements are likely to be a lower bound of the effects of budgetary restraint on decisions in the sense that budgetary constraints may impact decisions throughout the fiscal year.

We find that the 2011-reform that introduced fiscal disciplinary devices contributed to a decline in budget overruns in general, but especially for out-of-home care. The result that the budget for out-of-home care is particularly sensitive to budget restrictions suggests that it is easier to up- or downscale child protective services with respect to other activities. One explanation for this could be that child protective services only affect a small and often marginalized group and are not subject to the same kind of public attention as core welfare activities such as the universal provision of schools and daycare. Another explanation for why expenditure on child protective services are particularly sensitive to budget variation can be attributed to the organization of the services in this area. While budgets for daycare, schooling or elderly care rely heavily on short-term fixed costs for buildings and wages for employees in fixed positions, spending on out-of-home care consists of more variable costs. Municipalities hire foster families on short-term contracts, and care interventions can be changed or terminated at a short notice. Thus, activities related to out-of-home care are relatively easy to scale up or down in case of budgetary pressure. The problem may be particularly relevant for small municipalities with fewer options for cost smoothing, where a few additional children in out-of-home care can lead to a substantial budget overrun. However, we do not find any significant difference in the estimated effects across municipalities depending on their population size, the demographic composition or resource pressure given by demographic composition (S5 and S6 Tables in S1 Appendix).

Our quantitative results are in accordance with qualitative evidence suggesting that financial circumstances and public expenditure aspects are present in discussions among municipal caseworkers when making child protection decisions. Qualitative evidence based on interviews with municipal decision makers indicates that managers do pay close attention to the budget, but budgetary issues seem to play a minor role in serious cases, for example involving violence and abuse [29]. This suggests that budget considerations primarily affect the marginal child protection case when there is doubt as to whether an out-of-home placement is necessary. Recent Danish media coverage has furthermore documented cases in which municipalities ordered caseworkers to find considerable cost reductions on child protection cases [30].

Our research demonstrates that imposing sanctions on local municipalities for budget overruns can have unintended and potentially harmful consequences for the provision of welfare services, even when the sanction applies to the total budget. While our paper does not question the appropriateness of national levels of expenditure for out-of-home care or other child protective measures, we raise an important policy question regarding how to ensure that policies designed to ensure fiscal stability do not jeopardize vital welfare tasks such as protecting at-risk children. In particular, our analyses suggest that important decisions regarding child protection may be impacted by local budget concerns. Policy makers should thus consider

unintended side effects when designing disciplinary budget devices. Our results underline the need to carefully consider if sufficient public provision of essential welfare services can be guaranteed when delegated to the local government level, or whether more centralized coordination is warranted. Our paper thus contributes to the fiscal federalism literature by directing attention towards the fact that the public sector faces difficult trade-offs between providing effective and credible measures to curb local government expenditure and ensuring safety and equal opportunities for vulnerable children.

## Supporting information

**S1 Appendix.**
(DOCX)

## Acknowledgments

We would like to thank Joseph Doyle for inspiring suggestions and discussions, and Julie Uldum Lyhr for invaluable research assistance. We further thank Kurt Houlberg, VIVE, for explaining the municipal budget structure and for providing us with data on resource pressure at the municipal level. We also thank participants in the Children in Care workshop at the Royal Holloway University, at seminars at Stockholm School of Economics and CEBI, at the European Economic Association (EEA) annual conference in 2020, and at the 20th Journées Louis-André Gérard-Varet (LAGV) conference in 2021 for helpful comments and suggestions.

## Author Contributions

**Conceptualization:** Mette Ejrnæs, Mette Gørtz.

**Data curation:** Petra Gram Cavalca, Mette Ejrnæs.

**Formal analysis:** Petra Gram Cavalca, Mette Ejrnæs, Mette Gørtz.

**Funding acquisition:** Mette Ejrnæs, Mette Gørtz.

**Investigation:** Petra Gram Cavalca, Mette Ejrnæs, Mette Gørtz.

**Methodology:** Petra Gram Cavalca, Mette Ejrnæs, Mette Gørtz.

**Project administration:** Mette Ejrnæs.

**Supervision:** Mette Ejrnæs, Mette Gørtz.

**Validation:** Petra Gram Cavalca.

**Writing – original draft:** Petra Gram Cavalca, Mette Ejrnæs, Mette Gørtz.

**Writing – review & editing:** Petra Gram Cavalca, Mette Ejrnæs, Mette Gørtz.

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
