## [Decision Letter · Decision Letter 0]

19 Apr 2021

PONE-D-21-06271

Trading off fiscal budget adherence and child protection

PLOS ONE

Dear Dr. Gørtz,

Thank you for submitting your manuscript to PLOS ONE. After careful consideration, we feel that it has merit but does not fully meet PLOS ONE’s publication criteria as it currently stands. Therefore, we invite you to submit a revised version of the manuscript that addresses the points raised during the review process.

All three referees find the general question to be interesting but Reviewers 1 and 3 raise a number of important concerns.

The first major concern is related to the lack of any evidence that reducing spending leads to adverse effects on children. Such evidence and discussion should be included in the revised version as well as clarifying the theoretical framework as suggested by Reviewer 1. 

The second major conern has to do with the empirical strategy which is not always clearly presented and the discussion of the findings is sometimes incomplete. Reviewer 1 suggests an alternative way of assigning municipalities into low or high risk of exceeding their budget based on their budget history before the reform of 2011, which is worth exploring. Reviewer 3 is concerned with the individual-level analysis pointing correctly that there is no variation across individuals within a municipality. Indeed, equation (2) seems to suggest that the budget share of municipality (k) varies at the level of individual (i), which is not the case. As a result the analysis could easily be focused at the municipality level unless a convincing argument can be made for the individual-level analysis. Reviewer 2 suggest to motivate and consider different cutoffs than the 60% used.

All reviewers also make several other minor suggestions which can help to improve the paper and should be relatively easy to address. 

We look forward to receiving your revised manuscript.

Kind regards,

Konstantinos Tatsiramos

Academic Editor

PLOS ONE

Journal Requirements:

3a) If there are ethical or legal restrictions on sharing a de-identified data set, please explain them in detail (e.g., data contain potentially sensitive information, data are owned by a third-party organization, etc.) and who has imposed them (e.g., an ethics committee). Please also provide contact information for a data access committee, ethics committee, or other institutional body to which data requests may be sent.

3b) If there are no restrictions, please upload the minimal anonymized data set necessary to replicate your study findings as either Supporting Information files or to a stable, public repository and provide us with the relevant URLs, DOIs, or accession numbers. For a list of acceptable repositories, please see http://journals.plos.org/plosone/s/data-availability#loc-recommended-repositories.

4. We note that Figures A4 and A7 in your submission contain map images which may be copyrighted. All PLOS content is published under the Creative Commons Attribution License (CC BY 4.0), which means that the manuscript, images, and Supporting Information files will be freely available online, and any third party is permitted to access, download, copy, distribute, and use these materials in any way, even commercially, with proper attribution. For these reasons, we cannot publish previously copyrighted maps or satellite images created using proprietary data, such as Google software (Google Maps, Street View, and Earth). For more information, see our copyright guidelines: http://journals.plos.org/plosone/s/licenses-and-copyright.

1.              You may seek permission from the original copyright holder of Figures A4 and A7 to publish the content specifically under the CC BY 4.0 license. 

Reviewers' comments:

Reviewer's Responses to Questions

**Comments to the Author**

1. Is the manuscript technically sound, and do the data support the conclusions?

Reviewer #1: No

Reviewer #2: Yes

Reviewer #3: Partly

2. Has the statistical analysis been performed appropriately and rigorously? 

Reviewer #1: N/A

Reviewer #2: Yes

Reviewer #3: Yes

3. Have the authors made all data underlying the findings in their manuscript fully available?

Reviewer #1: No

Reviewer #2: No

Reviewer #3: No

4. Is the manuscript presented in an intelligible fashion and written in standard English?

Reviewer #1: Yes

Reviewer #2: Yes

Reviewer #3: Yes

5. Review Comments to the Author

Reviewer #1: This paper addresses the link between fiscal constraints and investments/management of child protection programs, using danish data. The underlying idea is that, if you impose fiscal constraints to lower levels of government, you might end up having side effects, as the ones detected on the allocation of resources for child protection. Having granular data. the authors can disentangle the use of money in the first part of the year, and they exploit the differences between those municipalities having spent the majority of their budget in child protection by July 1st (high spending) and those not having spent the majority of their budget in child protection by July 1st (low spending), to assess the role of budget constraints introduced starting 2011 in Denmark.

While the topic is extremely interesting, I find several conclusions reached by the authors as unsubstantiated and therefore needy of more support. I provide some suggestions in this regard, not in the order of importance.

\\item I found the takeaway of the paper a little bit naive in terms of policy implications. It basically proposes that there should be no budget constraints and municipality should be set free to overrun any expenditure limits. Yet, even if this should be the case, resources are scarce, even waiving the EU stability pact. This means that choices would be needed on which item to allocate the scarce financial resources available in the municipal budget. At that point we would need a criterion to prioritize resources allocation, which does not emerge by the authors' analysis. Another scenario could be that municipalities could raise their own taxes to increase revenues so that they could fuel child protection vis a vis other welfare programs. However, this scenario should address the possibility to "vote with the feet", which stands for within a country migration. Hence, I do think that the paper would substantially benefit from a clear theoretical framework at the beginning spelling out which kind of institutional setting the authors have in mind.

\\item Related to the previous point and the policy implications: the claim at the end of the paper ``potentially harmful consequences" is not proven. The authors show that there is a decrease in certain actions related to child-protection programs, but they do not bring any evidence that this reduction is associated to an increase in social distress for the part of the population most interested by the program. Are high school drop-outs in those municipalities increasing? Is the rate of police interventions for child neglecting increasing? Are there more emergency room recoveries for the population below 18? I do think this is pivotal to assess the relevance at the society level of these cuts or reduction in the use of child protection measures.

\\item A final remark on policy implications: the difference between day care and child protection is not just a matter of the structure of the costs (as suggested at p.26) but it is also a matter of the proportion of residents affected by the service. Using the descriptive stats, the authors should provide a flavor of how relevant are the population groups more likely to be interested by child protection in a given municipality in a given year. For instance, how relevant are single mums in a certain municipality, or immigrants? They could use these infos to run specific heterogeneities.

\\item The measure of taking kids out of their home is an extreme ratio. There might be early intervention programs in order to avoid this extreme measure. Has anything happen in that respect during the observational period? It appears that there has been a decrease in the use of out-of-home procedure independent from the 2011 reform (see Figure A5), could it be due to the use of alternative solutions?

\\item As for the empirical approach, why do not adopt a more simple scenario. Split municipalities--using an aggregated measure referred to the period before the reform-- into more and less fiscal conservative (so to say, more or less available to run over expenditures). For instance using the average excess budget run in the pre-reform period. Then you know that the introduction of the 2011 reform will have a different intensity, and you can check the effect of the intensity of the treatment on child-protection actions and expenditures. I find the use of the expenditures up to July 1st as an extra, rather than an intuitive baseline.

\\item As an extra heterogeneities I would consider the alignment between the local and the national government (i.e. more funding or more flexibility), and to consider the distance from the election month, which could affect spending decisions (conditional to the relevance of the affected population).

Reviewer #2: Please see attached PDF file. This is requiring a minimum number of characters in this box to move on to the next screen, so I am adding some additional characters here so I can submit this review on the next page.

Reviewer #3: This paper investigates an interesting question at the intersection of local public finance and youth welfare by searching for the effects of tightening budgets within any given year on expenditures on child protection (and especially fostering) in Danish municipalities. It utilizes the very rich Danish registry data that offers interesting detail on this very specific issue. The authors argue that tight budgets lead to a failure to protect vulnerable youth.

1. Although the authors place these results into a normative framework, ex ante it is not entirely clear whether spending more on various forms of child protection also brings adequate individual benefits. It is very easy to imagine narratives where this is the case, but it may support the normative statements better if the authors could give more background on the necessity and benefits of the different types of expenditures. E.g., how relevant or how much needed is out-of-home care for youth above 18 (which seems to be the most affected cohort by budgetary tightness); when this support is getting cut, is it a reasonable cost saving strategy or a government failure? The authors could help the readers to assess this by arguing about the relative benefits of different forms of care and types of expenditures more explicitly. All this may seem obvious, but stating it clearly would be helpful.

2. Municipal-level strategy:

- Are budget shares based on an ex-post view so that they always amount to 1 by the end of the year? Or are they calculated in reference to the original municipal budgets, so that some municipalities may actually decide to overdraw (or undercut) their yearly budget on care at the costs (or benefits) of something else? Although this is described, I was still not sure which of the two is correct (providing basic descriptive statistics would have helped). If the former, does this affect the empirical strategy in any ways?

- Model 2 in table 1 shows insignificant interaction effects. But would this not invalidate the authors' argument that all these problems became more relevant after the new budgetary rules were introduced? At the beginning there are some arguments in this direction, but then there is virtually no discussion of whether this result invalidates the previous discussion.

- Column 3 of table 1: results are shown but not discussed. A non-attentive reader could miss entirely the meaning of this column, especially if only looking at the table.

3. Individual-level strategy:

I had difficulties to clearly see the estimation strategy. Whereas the municipality-level regressions are fairly clear, I was somewhat confused by the individual-level estimates.

- First of all, I do not clearly see why are those estimates run at the individual level at all. The main variation is at the municipality-month level. Could not aggregate series be generated on the flows of children (number of below-18 ending placement, number of above-18 ending placement), etc.? I see that individual-level regressions offer the possibility to control for individual and household characteristics, but I doubt that these would change any of the results and many of the characteristics could also be accounted for by an individual fixed effect. Moreover, when looking at the individual-level estimations, it seems to me that some of these outcomes beg for a duration model, e.g. when referring to the likelihood of ending a spell. The individual-level results seem to be run at very different sample sizes, again raising the question whether sample selection would be relevant in any form here. Such concerns would be eliminated when looking at monthly flow data at the municipality level w/o losing any of the narrative content?

- It is not clear to me why are year and month effects always listed differently. I would have thought that all models include simply time (which is here month-year) fixed effects. The way year and month dummies are separately emphasized, it seems like there are T year dummies, and 12 year-of-month dummies to account for seasonality. Instead, I would add Tx12 month FEs. Maybe this is what happening here, it could be then described as monthly FEs.

- From the individual-level regressions only selected marginal effects are plotted. As the results are not shown in the main paper, it is hard to imagine, how exactly these results are estimated (e.g., how the interaction effects of before 2011 and after enter the model)? I also did not understand why would these interaction effects refer to July (stated on line 348). In a monthly panel, the marginal effects average out municipalities with above 60% budget in any of the months?

Smaller comments:

- The tables should clearly state the dependent variables used.

- The description of the estimating equation (2) would be clearer if it would include all nested fixed effects (and always in the same order): y_ikt, B_ik

t-1, etc.

- There are a few typos: e.g., line 229.

6. PLOS authors have the option to publish the peer review history of their article (what does this mean?). If published, this will include your full peer review and any attached files.

Reviewer #1: No

Reviewer #2: No

Reviewer #3: No

---

## [Author Response · Author response to Decision Letter 0]

6 Sep 2021

Dear Editor and Reviewer # 1, 2 and 3,

Thank you for the constructive and insightful comments and suggestions, which we address in the enclosed Response to reviewer.

Best,

Mette Gørtz

---

## [Decision Letter · Decision Letter 1]

19 Nov 2021

PONE-D-21-06271R1Trading off fiscal budget adherence and child protectionPLOS ONE

Dear Dr. Gørtz,

Thank you for submitting your manuscript to PLOS ONE. I have received comments from the reviewers on the revised article and there are only a couple of comments raised by R1 and R2 that I would like to invite you to consider in another minor revision.

We look forward to receiving your revised manuscript.

Kind regards,

Konstantinos Tatsiramos

Academic Editor

PLOS ONE

Journal Requirements:

Reviewers' comments:

Reviewer's Responses to Questions

**Comments to the Author**

1. If the authors have adequately addressed your comments raised in a previous round of review and you feel that this manuscript is now acceptable for publication, you may indicate that here to bypass the “Comments to the Author” section, enter your conflict of interest statement in the “Confidential to Editor” section, and submit your "Accept" recommendation.

Reviewer #1: (No Response)

Reviewer #2: (No Response)

Reviewer #3: (No Response)

2. Is the manuscript technically sound, and do the data support the conclusions?

Reviewer #1: (No Response)

Reviewer #2: Yes

Reviewer #3: Partly

3. Has the statistical analysis been performed appropriately and rigorously? 

Reviewer #1: (No Response)

Reviewer #2: Yes

Reviewer #3: Yes

4. Have the authors made all data underlying the findings in their manuscript fully available?

Reviewer #1: (No Response)

Reviewer #2: No

Reviewer #3: No

5. Is the manuscript presented in an intelligible fashion and written in standard English?

Reviewer #1: Yes

Reviewer #2: Yes

Reviewer #3: Yes

6. Review Comments to the Author

Reviewer #1: See attached file where I inserted my comments fort he aiuthors. I think they have to slightly adjust the selling point

Reviewer #2: The authors have successfully addressed almost all of my concerns. I just think one minor change is necessary - when describing the individual-level analysis in Section 3.3, it needs to be clear that for 4) we are interested in the probability of receiving an out-of-home placement rather than preventive action (right now, it reads like it is comparing receiving either of these to receiving nothing). At the moment, that is only clear by looking in the Appendix. Thanks!

Reviewer #3: The argumentation of the paper became substantially clearer, thank you. On some modelling choices I do not agree with the authors but I'm not convinced that the academic publications process should lead to complete agreement, so that's fine.

7. PLOS authors have the option to publish the peer review history of their article (what does this mean?). If published, this will include your full peer review and any attached files.

Reviewer #1: No

Reviewer #2: No

Reviewer #3: No

---

## [Author Response · Author response to Decision Letter 1]

4 Dec 2021

Our response to the editor and reviewers can be found below (and also in the attached cover letter and response letter).

Reviewer #1

I appreciate the attempts of the authors to address all the raise concerns. Yet I do think they have to turn down the selling point on “We thus highlight that enforcing strict budget adherence can have unintended and potentially devastating side effects." They do show that there is a trade off and that there is a decrease in the outcome of interest, but they do not have a smoking gun on the cognitive performance or emotional response in the treated children. I would suggest only to be open to that in their narrative.

Response: We have now downplayed this sentence so that we abstain from talking about possible effects on child outcomes. Instead, we write in the abstract, on the bottom of page 1, that:

“We thus highlight that enforcing strict budget adherence may be in conflict with social policy goals.”

Reviewer #2

The authors have successfully addressed almost all of my concerns. I just think one minor change is necessary - when describing the individual-level analysis in Section 3.3, it needs to be clear that for 4) we are interested in the probability of receiving an out-of-home placement rather than preventive action (right now, it reads like it is comparing receiving either of these to receiving nothing). At the moment, that is only clear by looking in the Appendix. Thanks!

Response: We have clarified this by changing the formulation in section 3.3 (bottom of page 14) to:

“4) initiating a preventive care measure rather than an out-of-home placement”.

---

## [Editor Report · Decision Letter 2]

9 Dec 2021

Trading off fiscal budget adherence and child protection

PONE-D-21-06271R2

Dear Dr. Gørtz,

We’re pleased to inform you that your manuscript has been judged scientifically suitable for publication and will be formally accepted for publication once it meets all outstanding technical requirements.

Kind regards,

Konstantinos Tatsiramos

Academic Editor

PLOS ONE

---

## [Editor Report · Acceptance letter]

11 Mar 2022

PONE-D-21-06271R2 

Trading off fiscal budget adherence and child protection 

Dear Dr. Gørtz:

I'm pleased to inform you that your manuscript has been deemed suitable for publication in PLOS ONE. Congratulations! Your manuscript is now with our production department. 

Kind regards, 

on behalf of

Prof. Konstantinos Tatsiramos 

Academic Editor

PLOS ONE